# Reduction of Li$^+$ within a borate anion

Haokun Li [1], Jiachen Yao [1], Gan Xu [1], Shek-Man Yiu[1], Chi-Kit Siu [1], Zhen Wang [1], Yung-Kang Peng[1], Yi Xie [2], Ying Wang [2] & Zhenpin Lu [1] ✉

Group 1 elements exhibit the lowest electronegativity values in the Periodic Table. The chemical reduction of Group 1 metal cations M$^+$ to M(0) is extremely challenging. Common tetraaryl borates demonstrate limited redox properties and are prone to decomposition upon oxidation. In this study, by employing simple yet versatile bipyridines as ligands, we synthesized a series of redox-active borate anions characterized by NMR and X-ray single-crystal diffraction. Notably, the borate anion can realize the reduction of Li$^+$, generating elemental lithium metal and boron radical, thereby demonstrating its potent reducing ability. Furthermore, it can serve as a powerful two-electron-reducing reagent and be readily applied in various reductive homo-coupling reactions and Birch reduction of acridine. Additionally, this borate anion demonstrates its catalytic ability in the selective two-electron reduction of CO$_2$ into CO.

In 1947, Wittig and coworkers reported the first example of tetraphenylborate, LiB(C$_6$H$_5$)$_4$, **1**[1,2]. Other prominent examples of tetraphenylborates include its fluorinated congeners, such as [B(C$_6$F$_5$)$_4$]$^-$ and [B(C$_6$H$_3$(CF$_3$)$_2$)$_4$]$^-$, **2**–**3** (Fig. 1a)[3]. These tetraaryl borates are known as weak coordinating anions[4] and have found applications in various fields, including organic synthesis[5,6], coordination chemistry[7–9], material science[10–12], and biomedical studies[13–15].

In general, these tetraaryl borate anions are stable under ambient air conditions. However, they are not redox-active species since the corresponding boron radical and cation species are unstable and decompose under oxidizing conditions, leading to the formation of biphenyl products (Fig. 1b)[5,16]. We envision that the installation of a non-innocent ligand at the boron atom may endow the corresponding borate anion with interesting redox properties.

Bipyridine, a readily available and versatile reagent, has been widely employed as a non-innocent ligand for transition metals in coordination chemistry[17,18], supramolecular chemistry[19,20], and catalysis[21]. Previously, Wagner and co-workers reported a series of 2,2′-bipyridylboronium compounds, which can undergo two-electron reduction as confirmed by electrochemistry[22–24]. These results suggest the promising redox properties of bipyridine-coordinated borate anions. Although bipyridine-stabilized boronium and boron radical compounds were successfully isolated[22,25], the corresponding borate anion was not reported (Fig. 1c).

The elements in Group 1 of the Periodic Table are characterized by having the lowest electronegativity values. The chemical reduction of Group-1 metal cations M$^+$ to form their corresponding zero-valent species M(0) is exceptionally rare and represents one of the most challenging endeavors in the field of synthetic inorganic chemistry, compared to electrochemical and photochemical regimes. One major hurdle is the absence of a suitable reducing agent capable of surpassing the highly negative redox potential. Recently, the reduction of Na$^+$ to zero-valent Na metal was achieved through low-valent magnesium species reported by the Harder group[26] and the Hill/McMullin groups[27]. Lu and co-workers realized a selective reduction of Li$^+$/K$^+$ in the heterobimetallic electride[28]. The Hill/McMullin groups reported that the reduction of Li$^+$/K$^+$ can be achieved by other heavier group 1 elements within chloroberyllate compounds[29]. In these reactions, the reduction of Group-1 metal cations was realized through metal-containing systems with special ligands and complicated/harsh synthetic approaches.

Herein, we report the first example of bipyridine-coordinated borate anions (Fig. 1d). Notably, the borate anion shows a robust reduction ability and can realize a reduction of Li$^+$ into the corresponding elemental metallic species, forming the boron radical. Our results demonstrate that the choice of solvent is crucial for the reduction of Li$^+$: etherate solvents, such as THF and crown ether, can stabilize the Li$^+$; in toluene, the borate anion can promote the

[1]Department of Chemistry, State Key Laboratory of Marine Pollution, City University of Hong Kong, Kowloon Tong, Hong Kong SAR, P. R. China. [2]Department of Chemistry, The Chinese University of Hong Kong, Shatin, Hong Kong SAR, P. R. China. ✉e-mail: zhenpilu@cityu.edu.hk

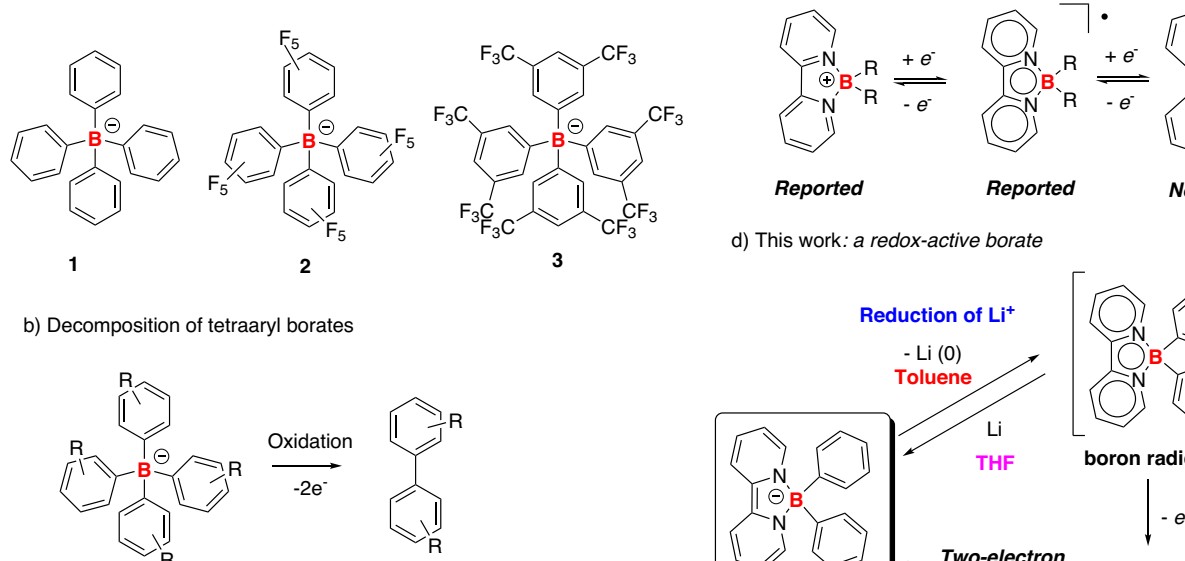

a) Commonly used tetraphenyl borate anions

b) Decomposition of tetraaryl borates

c) Bipyridine stabilized organoboron species

d) This work: a redox-active borate

**Fig. 1 | Examples and reactivity of tetraaryl borates. a** Commonly used tetraphenyl borate anions; **b** decomposition of tetraaryl borates; **c** bipyridine stabilized boron species; **d** the borate anion reported in this study.

formation of Li(0) and corresponding boron radical. Furthermore, the borate anion can undergo two-electron transfer to generate the corresponding boronium, which can be readily applied in reductive heteroatom-coupling reactions, pyridine coupling, the reduction of acridine, and catalytic two-electron reduction of $CO_2$ into CO.

## Results

### Synthesis, characterization, and reactivity of borate anion

The borate anion **5** was synthesized through the direct reaction of PhLi and a known organoboron species **4**[30,31] (Fig. 2), the synthesis of which can be achieved through a one-pot fashion in 67% isolated yield on a gram scale. In the [11]B NMR spectra, **5** exhibits a signal at 2.79 ppm, in agreement with the presence of a tetracoordinate boron center. The [1]H NMR spectra show four signals at 3.82, 4.64, 5.11, and 5.68 ppm, attributed to the protons from the bipyridine rings, indicating the dearomatization of bipyridine moieties in **5**. The identity of **5** was unambiguously confirmed by X-ray single-crystal studies. The lithium cation was coordinated with four THF molecules in the solid state (Fig. 3, left). Similarly, the reduced form of bipyridine at the coordination sphere of early transition metals has been reported[32–37].

The borate anion **5** was stable in the THF solution. When the solvent was changed to toluene, **5** was gradually converted to an NMR silent species **6** and a black precipitate. Compound **6** was isolated in 85% yields (Fig. 2) and characterized by single-crystal X-ray analyses, revealing the presence of a tetra-coordinated boron center similar to borate anion **5** (Fig. 3, right). Since compound **6** is a neutral species, we infer that it should be a radical species. The radical **6** was further analyzed by EPR spectra in a toluene solution at room temperature. An EPR signal (centered at $g_{iso} = 2.0036$) was observed (Fig. 4). The experimental data were then compared with the simulated one, and the hyperfine

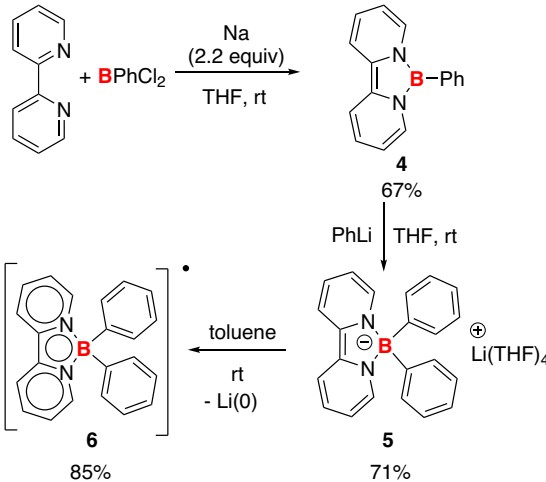

**Fig. 2 | Synthesis of borate anion 5 and boron radical 6. 5** was synthesized from compound **4** and PhLi, and the decomposition of **5** in toluene generated **6** and metallic lithium.

splitting $(a(^{10}B, 19.9\%) = 0.05\,G,\ a(^{11}B, 80.1\%) = 0.16\,G,$ $a(^{14}N) = 3.57\,G,\ a(^{1}H) = 1.01, 7.77, 8.89, 10.77\,G)$ suggests that the spin density is mainly delocalized in the bipyridine and $BN_2C_2$ rings.

Besides, it is crucial to identify the lithium-containing species in the product during the formation of radical **6**. Since compound **6** was isolated in a high yield (85%), and no other side products were traced by NMR spectra. We speculate that lithium cations might be converted into Li(0) species. To verify our hypothesis, the black

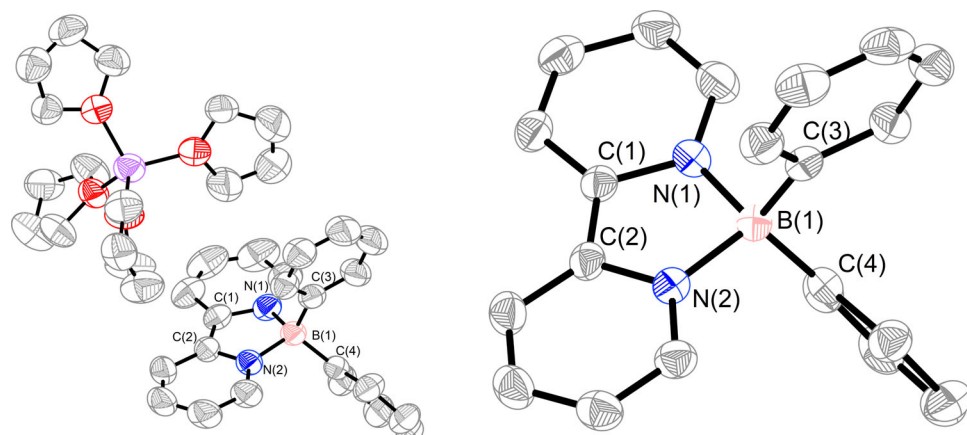

**Fig. 3 | Molecular structure of borate anion 5 (left) and boron radical 6 (right)** (thermal ellipsoids are set at the 50% probability level, and all hydrogen atoms are omitted for clarity). Selected bond distances (Å) of compound **5**: N1–B1 1.564(3), N2–B1 1.557(3), N1–C1 1.417(3), N2–C2 1.419(3), C1–C2 1.359(3); selected bond distances (Å) of compound **6**: N1–B1 1.590(3), N2–B1 1.578(3), N1–C1 1.386(3), N2–C2 1.386(3), C1–C2 1.404(3).

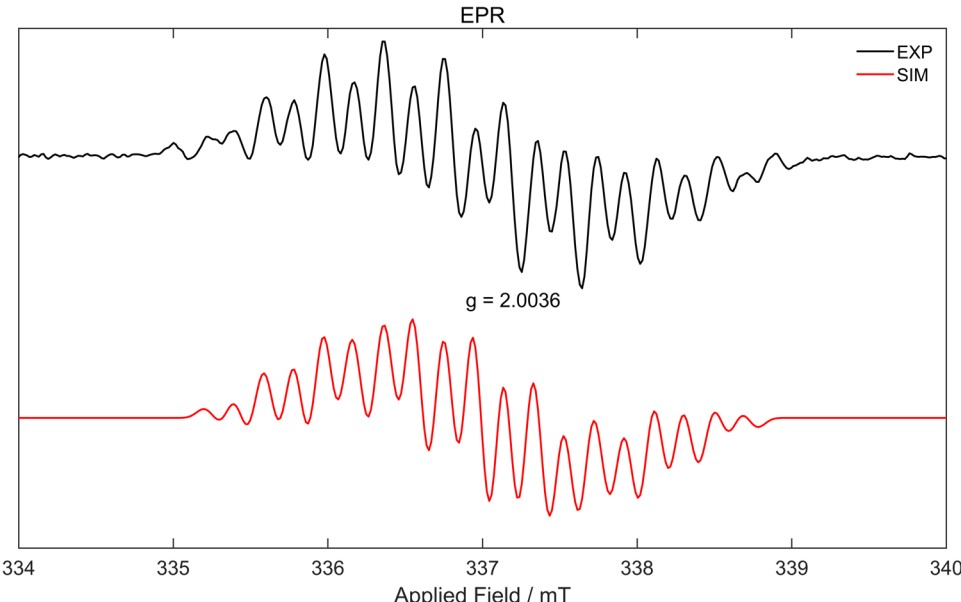

**Fig. 4 | Experimental (black) and simulated (red) EPR spectra of** 6 **in toluene at room temperature.** The simulation was performed on two isotopomers, ($^{11}$B)/($^{10}$B), in a 4:1 ratio.

precipitate generated from the reaction was collected, and the X-ray photoelectron spectroscopy (XPS) data of the lithium products showed a signal at 54.58 eV, which is close to that of commercial lithium samples at 54.38 eV (Fig. 5). X-ray powder diffraction (see Supplementary Fig. 39) and inductively coupled plasma-optical emission spectroscopy (ICP-OES) were also utilized to characterize the products (see Supplementary information). These results clearly confirmed the presence of lithium (0) species in the products. Furthermore, the reduction of lithium cation was supported by computation studies, showing that such a redox process is thermodynamically favored ($\Delta G = -51.82$ kcal/mol) (see the Supplementary information). To the best of our knowledge, this study represents a rare example of lithium cation reduction within an organic species.

When THF was added to the mixture of compound **6** and in-situ generated Li(0), borate anion **5** was clearly regenerated (Fig. 6a). (see Supplementary Fig. 28) Furthermore, when 12-crown-4 ether and compound **5** were mixed in toluene, no formation of compound **6** was observed (Fig. 6b), as confirmed by the EPR (see Supplementary Fig. 38), indicating that the coordination of THF or crown ether is crucial for the stabilization of lithium cations in **5**. Additionally, the reaction of compound **4** and PhLi in toluene generated compound **6** with a 94% yield (Fig. 6c). In these transformations, we have demonstrated that the choice of solvent can control the generation of boron radical or borate anion.

The redox properties of **6** were studied by cyclic voltammetry (CV) in THF solutions. A quasi-reversible process involving two reductions was observed in the CV spectra (Fig. 7). The second reduction occurs at −1.82 V vs. Fc/Fc$^+$, attributed to the reduced anionic species of radical **6**. Under electrochemical conditions, this borate anion can be oxidized to radical **6**, as shown in the first oxidation potential at −1.51 V; the second oxidation appears at −0.61 V, indicating the formation of boron cations; the reduction of boron cations at −0.91 V regenerates the radical **6**. Both the cationic and anionic derivatives of **6** were stable under electrochemical conditions (see Supplementary Fig. 35).

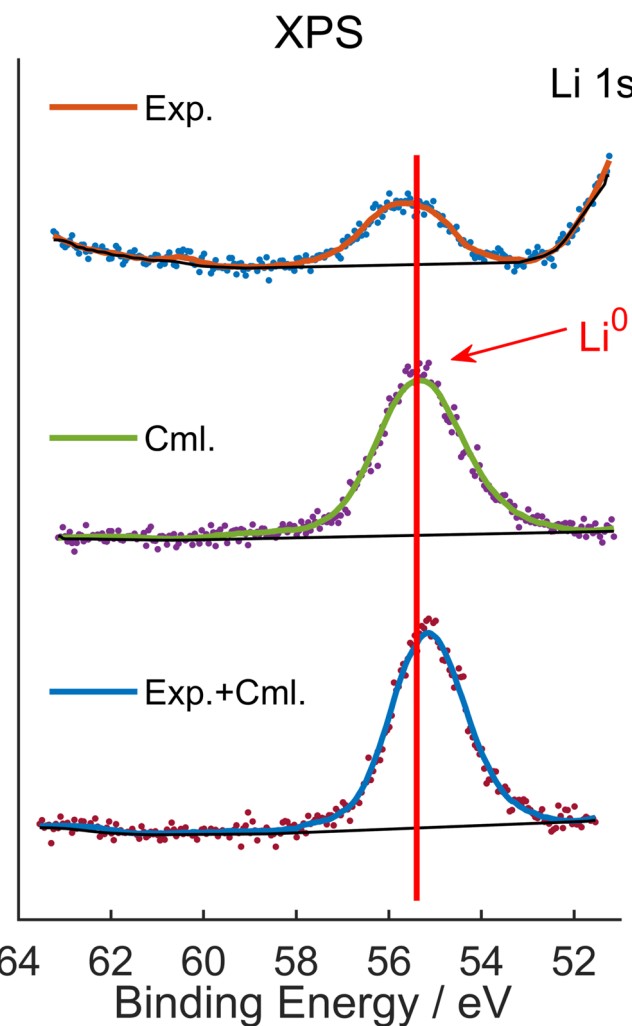

**Fig. 5 | XPS spectra of lithium-containing products (up), commercial metallic lithium (middle), and the mixture of our lithium-containing products/commercial metallic lithium (down).** XPS Analysis was acquired on Thermo Scientific K-Alpha.

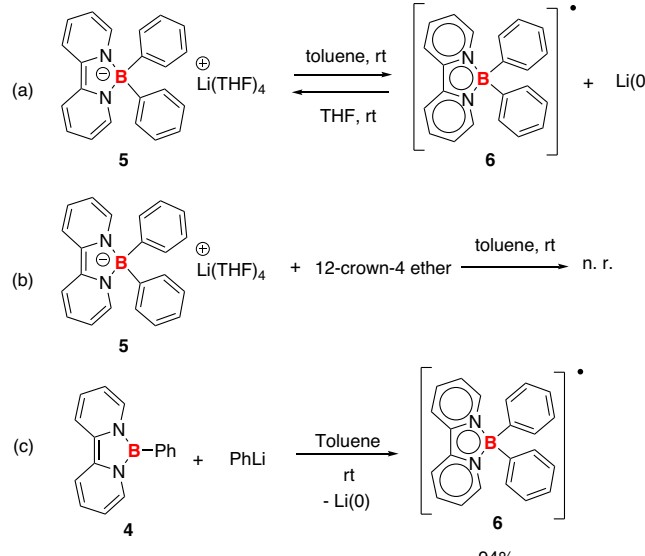

**Fig. 6 | Control experiments. a** The formation of **6** and the regeneration of **5**; **b** the reaction of **5** and 12-crown-4 ether in toluene; **c** synthesis of **6** from **4** and PhLi.

1 Å above the center of each ring (NICS(1))[38,39]. In general, NICS values of the two phenyl rings substituted at the boron atoms are negative and similar in these three species. In structures **5** and **6**, the two $C_5N$ rings are dearomatized with positive NICS(1) values, especially for the borate anion **5**, showing NICS(1) values of 25.89 and 25.92 (Fig. 11). In comparison, the two $C_5N$ rings in **7** are aromatic and exhibit negative NICS(1) values (−23.32). The extent of π-electron delocalization in **5**–**7** was further assessed through the analysis of current-induced density (ACID) anisotropy (see Supplementary Figs. 42–44). These results are in good agreement with their structural information.

Besides, DFT computation studies were performed to elucidate the electronic structure of compounds **5**, **6**, and **7**. The LUMOs of these compounds are all distributed on the bipyridine rings. For anion **5** and radical **6**, the bipyridine moieties also contribute mostly to the HOMOs, whereas the HOMO of boronium **7** is located on the two phenyl rings (see Supplementary Fig. 41). Moreover, anion **5** has the smallest HOMO−LUMO energy gap (4.18 eV) among the three derivatives (Fig. 12). These results indicate that the reactivity of compound **5** comes from the bipyridine moieties.

## Two-electron-transfer reactivity of borate anion 5'

Encouraged by the two-electron-transfer reactivity of **5'**, we explored the application of borate anion **5'** in reductive-coupling reactions. PPh₂Cl was chosen as the substrate to test the reactivity. The reaction of **5'** and PPh₂Cl (1:2) in THF solution was monitored at room temperature (Fig. 13). Indeed, a P-P coupling product, $P_2Ph_4$, was isolated as the final product, showing a signal at −15.01 ppm in the ³¹P NMR spectra[40–42]. The borate anion **5'** was converted to the boronium **8** in 80% isolated yield, confirming that compound **5'** acted as a two-electron-reducing agent in this reaction. Additionally, when a 1:1 mixture of the radical **6** and PPh₂Cl reacted in THF at room temperature, $P_2Ph_4$ was isolated in 86% yield. In comparison, when using sodium naphthalene as the reducing reagent, a 21% generation of $P_2Ph_4$ was observed based on the ³¹P NMR (see Supplementary Fig. 48).

Moreover, compound **5'** can also facilitate other element-element coupling reactions (Fig. 14). For example, nBu₃SnCl can be converted to nBu₃Sn−Sn(nBu)₃ in the presence of **5'** (Fig. 14a). The formation of the Sn−Sn bond in the final product was confirmed by the ¹¹⁹Sn-NMR spectra, showing a signal at −83.72 ppm attributed to the Sn−Sn

In agreement with the CV experiment, radical **6** can be reduced by potassium in THF, affording the borate anion **5'** in a 96% isolated yield (Fig. 8). Compound **5'** was fully characterized by NMR and HRMS spectra (see Supplementary Figs. 7–9). Contrary to lithium borate anion **5**, the potassium borate anion **5'** was stable in both THF and toluene, and no decomposition was observed at room temperature. However, when heating at 75 °C in toluene, **5'** was converted to radical **6** in a 10% isolated yield. The reduction of K⁺ into metallic potassium was confirmed by ICP-OES (see Supplementary information).

On the other hand, radical **6** can be oxidized to boronium **7** by an excess of CuCl in a THF solution (Fig. 9). Compound **7** was successfully characterized by NMR, HRMS spectra, and single-crystal X-ray analyses (Figs. 10 and Supplementary Figs. 10–12). The boron-containing frameworks of compounds **5**, **6**, and **7** bear similar geometries. However, the bond lengths of C1−C2 (1.359 Å), B1−N1 (1.564 Å), and B1−N2 (1.557 Å) in **5** are significantly shorter than those of **7** (1.469 Å, 1.601 Å, and 1.614 Å), revealing the non-aromatic feature of the two $C_5N$ rings. Furthermore, boronium **7** can be obtained through the oxidation of borate anion **5'**.

## Electronic structure of bipyridine-stabilized boron compounds

The aromatic characters of compounds **5**, **6**, and **7** were investigated through the nuclear independent chemical shift values at

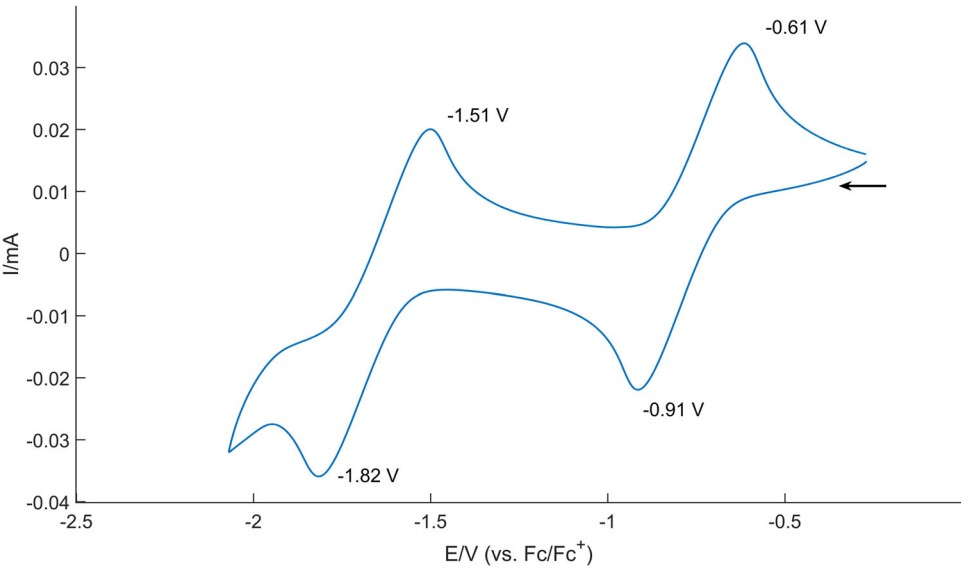

**Fig. 7 | Cyclic voltammograms (CVs) of 6** in THF solution containing 0.1 M [nBu₄N][PF₆]) at room temperature (scan rate: 100 mV/s). Ferrocene/ferrocenium couple was used as an internal standard.

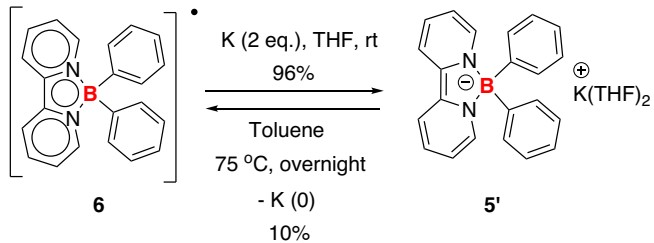

**Fig. 8 | Synthesis of borate anion 5' and boron radical 6.** The formation of **5'** or **6** can be controlled by different reaction conditions.

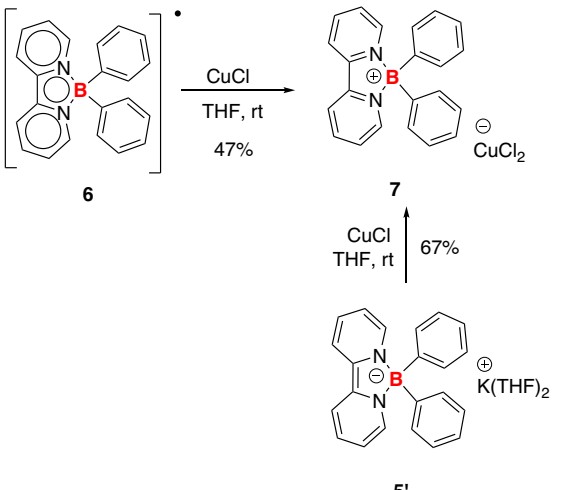

**Fig. 9 | Two different synthetic approaches to compound 7.** Either boron radical **6** or borate anion **5'** can undergo oxidation to give **7** in the presence of CuCl.

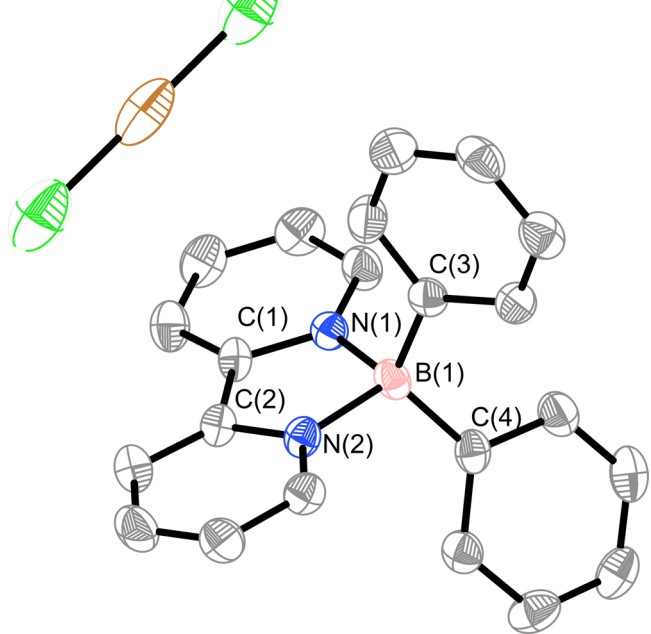

**Fig. 10 | Molecular structure of compound 7 (thermal ellipsoids are set at the 50% probability level, and all hydrogen atoms are omitted for clarity).** Selected bond distances (Å): N1–B1 1.614(6), N2–B1 1.601(6), N1–C1 1.352(6), N2–C2 1.346(6), C1–C2 1.469(6).

moieties (see Supplementary Fig. 20)[43]. In a similar manner, the formation of Se–Se and Ge–Ge can be realized (Fig. 14b, c). When the ammonium salt NHEt₃Cl reacted with **5'**, dihydrogen gas was released, confirmed by the online gas chromatography (see Supplementary Figs. 28 and 29) (Fig. 14d). A clean generation of NEt₃ was observed from the ¹H NMR spectra (see Supplementary Fig. 26). When using

sodium naphthalene as the reducing reagents in these reactions, the coupling products can not be obtained.

Additionally, compound **5'** can achieve reductive reactions on nitrogen-containing aromatic compounds. The reaction of pyridine and **5'** afforded the C–C coupling product, 4,4'-Bipyridine, in 24% isolated yield after quenching with methanol (Fig. 15). Recently, the Lu group realized a similar transformation using a highly reductive potassium-based electride reagent[44]. Furthermore, compound **5'** can promote Birch reduction of acridine, generating 9,10-dihydroacridine in a 58% isolated yield. These two reactions can't be realized using sodium naphthalene as a reducing reagent.

Finally, the two-electron reduction of $CO_2$ was also examined. The reaction of compound **5'** and an excess of $CO_2$ (1 atm) was conducted at room temperature in THF solution (Fig. 16a). This reaction generated $CO_3^{2-}$, confirmed by the $^{13}$C-NMR (see Supplementary Fig. 27), and CO as the final product, characterized by the online gas chromatography (see Supplementary Figs. 30–32). Notably, this reaction can be performed in a catalytic fashion, where 5 mol% **5'** was added to the potassium in THF with 1.2 equiv. $CO_2$ (1 atm), affording $K_2CO_3$ in 70% yield (Fig. 16b). In the absence of **5'**, potassium was barely consumed under $CO_2$ (1 atm) after one week (Fig. 16c). Similar transformations can also be realized in the presence of 9,10-dihydro-9,10-diboraanthracene anion and lithium, as reported by Wagner and co-workers[45]. Despite these achievements, our method represents a rare example of a transition-metal-free system for the selective reduction of $CO_2$ into CO[46–48].

In summary, we successfully synthesized a series of bipyridine-stabilized boron compounds, including borate anions, boron radicals, and boronium cations. The direct conversion of the anion **5** into radical **6** in a non-coordinating toluene solvent resulted in the reduction of $Li^+$ into the corresponding elemental metallic species, revealing the strong reducing ability of **5**. Furthermore, **5'** can mediate reductive homo-coupling reactions of organohalides, enabling the formation of P–P, Sn–Sn, Se–Se, and Ge–Ge bonds. Additionally, the direct reductive coupling of pyridine and the Birch reduction of acridine can be facilitated by **5'** in a similar fashion. More importantly, $CO_2$ can be catalytically converted to CO in the presence of potassium and 5 mol% of **5'**. Further study on the two-electron-transfer reactivity of these borate anions is currently underway.

## Data availability

All data generated or analyzed during this study are included in this manuscript (and its Supplementary Information). Details about materials and methods, experimental procedures, characterization data, and NMR spectra are available in the Supplementary Information. The X-ray crystallographic coordinates for structures reported in this study have been deposited at the Cambridge Crystallographic Data Centre (CCDC), under deposition numbers Deposition numbers CCDC 2301618 (for **5**), 2301617 (for **6**), and 2301616 (for **7**). These data can be obtained free of charge from The Cambridge Crystallographic Data Centre via www.ccdc.cam.ac.uk/data_request/cif. All data are also available from corresponding authors upon request. Source data are provided in this paper.

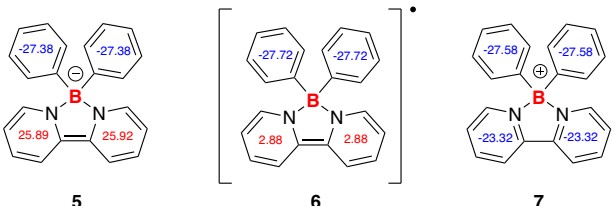

**Fig. 11 | NICS(1) values of compounds 5, 6, and 7.** Magnetic properties were calculated using the revTPSS/pcSseg-1 level of theory with optimized structures with the SMD solvent model (see Supplementary Information).

**LUMO** **HOMO**

0.77 eV  -3.41 eV

**Fig. 12 | Calculated LUMO and HOMO orbitals of anion 5.** All calculations were carried out using ORCA (version 5.0.4).

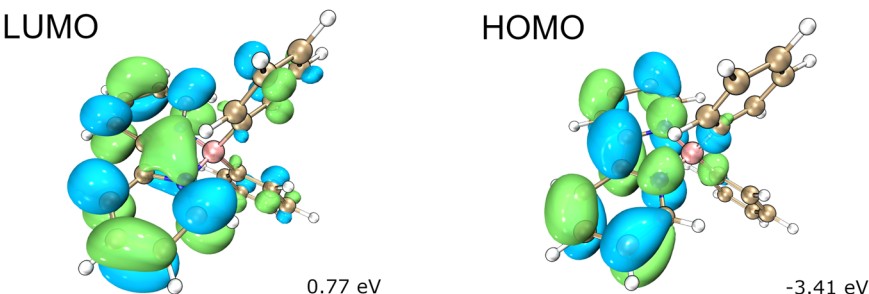

**Fig. 13 | Compound 5'/6 assisted P–P coupling reactions.** Both the borate anion **5'** and boron radical **6** were transformed into the boron cation **8**.

(a) $nBu_3SnCl \xrightarrow[\substack{THF, rt \\ -\ 8}]{0.5\ eq.\ 5'} nBu_3Sn\text{-}Sn(nBu)_3$

88%

(b) $PhSeCl \xrightarrow[\substack{THF, rt \\ -\ 8}]{0.5\ eq.\ 5'} PhSe\text{-}SePh$

71%

(c) $Et_3GeCl \xrightarrow[\substack{THF, rt \\ -\ 8}]{0.5\ eq.\ 5'} Et_3Ge\text{-}GeEt_3$

70%

(d) $NHEt_3Cl \xrightarrow[\substack{THF, rt \\ -\ 8}]{0.5\ eq.\ 5'} H_2\ +\ NEt_3$

>95% (NMR)

**Fig. 14 | Compound 5' assisted element-element coupling reactions. a** Sn–Sn coupling; **b** Se–Se coupling; **c** Ge–Ge coupling; **d** the formation of $H_2$.

**Fig. 15 | Direct C–C coupling of pyridine enabled by borated 5'; Birch reduction of acridine.**

(a) $CO_2 \xrightarrow[THF, rt]{5'} 1/2\ CO_3^{2-}\ +\ 1/2\ CO$

1 atm

(b) $CO_2 + K \xrightarrow[THF, rt]{5\%\ 5'} 1/2\ K_2CO_3\ +\ 1/2\ CO$

1 atm

70%

(c) $CO_2 + K \xrightarrow[THF, rt]{} n.r.$

1 atm

**Fig. 16 | $CO_2$ reduction. a** Reaction of **5'** and $CO_2$ (1 atm) in THF; **b** reaction of $CO_2$ (1 atm) and K in the presence of 5 mol% **5'** in THF; **c** reaction of $CO_2$ (1 atm) and K without **5'** in THF.

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

## Acknowledgements

This work was supported by grants from the Research Grants Council of The Hong Kong Special Administration Region (Project Nos. CityU 14303621 and 21310922) and start-up funds from the City University of Hong Kong (CityU). Z.L. thanks Prof. Huadong Wang (at Fudan University) and Prof. Zuowei Xie (at Southern University of Science and Technology) for their insightful discussions. C.K.S. thanks CityU for financial support (Project No. 7006003). The high-performance computing clusters of CityU ("*CityU Burgundy*") are acknowledged.

## Author contributions

H.L. performed all the experiments and conducted the computational and simulation study. K.S.M.Y. tested and solved single-crystal structures. C.K.S. directed and reviewed the computational study. Y.J., X.G. and Z.W. prepared some of the starting materials. Y.X. and Y.W. conducted the GC analyses on detecting CO and H$_2$. Y.K.P. completed the EPR measurement for the radical. L.Z. conceived the ideas and designed and directed the research. All the authors revised the paper.

## Competing interests

The authors declare no competing interest.
