## [Peer Review File · Nature Communications]

Reduction of Li⁺ within A Borate AnionReviewers' Comments:

Reviewer #1:

Remarks to the Author:

As I appreciate the authors' effort in investigating the redox properties of a bipyridyl-supported borate anion, I am not convinced that this manuscript is suitable for publication in Nature Communications.

While the claim regarding the borate anion's ability to serve as a two-electron reductant for the reduction of a series of compounds is scientifically sound, the assertion that this borate anion can reduce a lithium cation is, in my opinion, scientifically implausible.

I do not question the presence of lithium metal in the toluene solution; however, the significant difference in reduction potential between Li^+ (approximately -3.4 V) and the oxidation potential of the borate anion (-1.6 V) raises serious doubts about the proposed mechanism. A mere solvation effect does not seem sufficient to explain the observed phenomenon. There must be an alternative pathway leading to the formation of lithium metal, assuming the presence of lithium metal is accurate.

Considering the substantial gap in reduction potentials, approximately 1.5 V, between Li^+ and the borate anion, the proposed electron reduction mechanism appears unlikely. If there is an alternative explanation for the formation of lithium metal, it should be thoroughly investigated and presented in the manuscript to enhance the scientific credibility of the findings.

In summary, I find it challenging to support the publication of this study in Nature Communications based on the concerns raised. I believe addressing these critical issues is essential before further consideration for publication.

Reviewer #2:

Remarks to the Author:

This is a very interesting and well-conceived study that adds to the recent flurry of reports that indicate the chemical redox stability of group 1 cations is not as immutable as presented in most textbooks.

In this case the focus is a lithium diphenyl borate in which boron is further ligated by a bipyridine dianion. Although this compound is stable in THF due to coordination of the Li^+ , dissolution in toluene induces spontaneous reduction to lithium metal and a boron-containing radical; both products are convincingly characterized, by XPS and by X-ray diffraction and EPR, respectively.

Further study of the radical product shows that potassium reduction yields the analogous potassium borate. Although this compound is stable at room temperature, in toluene it undergoes similar self reduction to potassium metal. Furthermore, the potential of this species to act as a two electron reductant, with formation of the boronium cation, is demonstrated through its oxidation with Cu(I) . This latter behaviour is then exploited in a sequence of reactions which lead either to reductive E-E coupling of several p-block halides, pyridine and acridine. The potential for such species to act as a platform for redox-based catalysis is also shown through the reduction of CO_2 to CO by potassium in the presence of 5 mol% the K borate.

This is striking and thought provoking study. While I can think of many questions and suggestions for further experiments (I'm excited to see how general this approach can be made, particularly through extension to alkyl borates and more electron rich bipy variants), the scope of the current work is

perfectly suited to publication in Nature Comm and I am, thus, happy to recommend its acceptance.

Minor points:

The introductory section should also make mention of the the recent study of group 1 reduction in Nature Comms 2023, 14, 8147, which will have appeared as this paper was nearing submission.

Page 3, line 68 should read 'The experimental data were...'

Page 4, Cation to Fig. 2 should read '...in a 4:1 ratio.'

Reviewer #3:

Remarks to the Author:

This paper was well-revised after submission to other journal where the present reviewer made comments and now it describes chemistry better. However, identification of the present work is still not sufficient, considering the previously reported chemistry of bipyridines were not adequately referred and discussion on aromaticity is completely wrong. Therefore, the reviewer does not agree with publishing this paper. Please find the comments.

1) The largest claim in this paper is reducing ability of compound 5. However, this phenomena comes from the redox property of bipyridine. The reduced form of bipyridine at the coordination sphere of early transition metals has been reported. These examples should be cited.

<https://pubs.acs.org/doi/10.1021/acs.inorgchem.6b02954>

<https://chemistry-europe.onlinelibrary.wiley.com/doi/10.1002/chem.202003465>

<https://pubs.acs.org/doi/10.1021/om900943m>

<https://www.sciencedirect.com/science/article/pii/S0022328X1000505X?via=ihub>

<https://pubs.acs.org/doi/10.1021/acs.inorgchem.7b01344>

<https://pubs.rsc.org/en/content/articlelanding/2018/SC/C7SC05238C>

Also, electron transfer from naphthalene radical anion has been used in organic synthesis for a long time. Such chemistry should be compared with reactivity of 5 and 6 in this paper, especially for Schemes 6-9.

2) As mentioned in the previous reviewing, the statement "It is found that the BN₂C₂ rings in 5 and 6 are aromatic with negative NICS(1) values, whereas the two C₅N rings are dearomatized with positive NICS(1) values, especially for the borate anion 5, showing NICS(1) values of 25.89 and 25.92 (Figure 6)." in page 7 is wrong, considering the BN₂C₂ ring is not planar. The authors should know the NICS values are chemical shift and it can be affected by a change in charge density. As the authors added ACID plot in SI after the previous reviewing process, however, there is no ring current over B-N bonds, suggesting BN₂C₂ ring is not aromatic.

Reviewer #1:

As I appreciate the authors' effort in investigating the redox properties of a bipyridyl-supported borate anion, I am not convinced that this manuscript is suitable for publication in Nature Communications.

While the claim regarding the borate anion's ability to serve as a two-electron reductant for the reduction of a series of compounds is scientifically sound, the assertion that this borate anion can reduce a lithium cation is, in my opinion, scientifically implausible.

I do not question the presence of lithium metal in the toluene solution; however, the significant difference in reduction potential between Li^+ (approximately -3.4 V) and the oxidation potential of the borate anion (-1.6 V) raises serious doubts about the proposed mechanism. A mere solvation effect does not seem sufficient to explain the observed phenomenon. There must be an alternative pathway leading to the formation of lithium metal, assuming the presence of lithium metal is accurate.

Considering the substantial gap in reduction potentials, approximately 1.5 V, between Li^+ and the borate anion, the proposed electron reduction mechanism appears unlikely. If there is an alternative explanation for the formation of lithium metal, it should be thoroughly investigated and presented in the manuscript to enhance the scientific credibility of the findings.

In summary, I find it challenging to support the publication of this study in Nature Communications based on the concerns raised. I believe addressing these critical issues is essential before further consideration for publication.

ANS: The reduction potential of lithium ions is highly dependent on solvents. In a water solution, hydration substantially enhanced the stability of the lithium cation, thus resulting in a very negative reduction potential of -3.04 V. On the other hand, in a non-coordinating solvent such as toluene, its reduction potential will decrease significantly. While we cannot find any reduction potential of lithium ions in toluene from the literature, we did a theoretical calculation of the redox process of $\text{Li}^+ + \text{borate anion} \rightarrow \text{Li(s)} + \text{boron radical}$, which indicated that the reduction of lithium is thermodynamically favored ($\Delta G = -51.82 \text{ kcal/mol}$).

The high reducing ability of bipyridine-coordinated complexes is not unprecedented, due to the well-known non-innocent nature of bipyridine. For example, it was reported that a uranium bipyridine complex has a reduction potential of -2.7 V, which was attributed to the presence of bipyridine ligands (Inorg. Chem. 2017, 56, 5, 2792–2800).

We have added the following sentences in the revised version of the manuscript and supporting information:

Page 4, the last paragraph: **Furthermore, the reduction of lithium cation was supported by density functional theory (DFT) studies (revDSD-PBEB86-D4/ma-def2-QZVP/M06-2X/def2-TZVP in toluene), showing that such a redox process is thermodynamically favored ($\Delta G = -51.82 \text{ kcal/mol}$) (see the ESI)**

Page 8, the first paragraph: **These results indicate that the reactivity of compound **5** comes from the bipyridine moieties.**

Supporting information page 63:

A combination of computational and experimental methods was utilized to study the reduction of lithium.

For the lithium decomposition reaction of compound **5**, total energy change could be obtained from the calculated Gibbs free energies, $G\{\text{THF}(\text{tol})\}$, $G\{[\text{B}](\text{tol})\}$, $G\{[\text{Li}^+(\text{THF})_4][\text{B}^-](\text{tol})\}$ and $G\{\text{Li}(\text{s})\}$. To reduce the calculation error of a single atom, lithium gas to solid thermodynamic data were taken from the experimental results^[17].

Geometry optimization was performed at the density functional theory (DFT) using M062X^[10] with a def2-TZVP^[11] basis set. Counterions were omitted during calculations. The solution, toluene, was modeled by the Conductor-like Polarizable Continuum Model (CPCM)^[12] in optimization. Harmonic vibrational analyses were carried out to confirm if the optimized structure was a local minimum structure and to provide zero-point vibrational energy corrections and thermal corrections to various thermodynamic properties. For higher accuracy of electronic energy, double hybrid DFT, and larger basis set, revDSD-PBEP86-D4^[18]/ma-def2-QZVP^[11] level of theory with optimized structures was carried out with the SMD solvent model.^[15]

#1 s ^{#1}	ΔG (kcal/mol) #2	ΔH (kcal/mol) #2
Single molecule	-26.21 (revDSD-PBEP86)	-17.78 (revDSD-PBEP86)
	1.24 (M06-2X)	-15.43 (M06-2X)
Complete ionization	-51.82 (revDSD-PBEP86)	-25.80 (revDSD-PBEP86)
	-31.21 (M06-2X)	-41.51 (M06-2X)

#1: It means how we considered the energy of Lithium borate anion in calculation. Single molecular means it was a single molecular ($[\text{Li}^+(\text{THF})_4][\text{B}^-]$) in which the borate part and the lithium part were not ionized in toluene. Complete ionization means it was calculated in two parts, the borate part ($[\text{B}^-]$) and the lithium part ($[\text{Li}^+(\text{THF})_4]$) in toluene.

#2: DFT method used for electronic energy calculations.

Detailed calculated energies for each molecule are listed below in Hartree(h).

Compound	G (M062X)	H(M062X)	G (revDSD-PBEP86)	H (revDSD-PBEP86)
THF(tol)	-232.3464071	-232.3129632	-231.4634135	-231.4299695
$[\text{Li}^+(\text{THF})_4][\text{B}^-](\text{tol})$	-1920.182959	-1920.069231	-1912.6061219	-1912.492394
$[\text{Li}^+(\text{THF})_4](\text{tol})$	-936.8466343	-936.7691602	-933.2591693	-933.1816953
[B](tol)	-983.2517569	-983.1863109	-979.2810932	-979.2156473
$[\text{B}^-](\text{tol})$	-983.3496272	-983.2835549	-979.3389644	-979.272892
Li(g)	-7.495325	-7.4976869	-7.4648688	-7.4672293

And $\Delta H \text{ Li}(\text{g} \rightarrow \text{s}) = -159.3 \text{ kJ/mol}$, $S \text{ Li}(\text{g}) = 138.782 \text{ J}/(\text{mol} \cdot \text{K})$ and $S \text{ Li}(\text{s}) = 29.12 \text{ J}/(\text{mol} \cdot \text{K})$ were obtained from the experimental results.¹⁷

The ΔG revDSD-PBEP86 Single molecule in toluene can be obtained:

$$\Delta G \text{ (kcal/mol)} = G [\text{B}](\text{tol}) + G \text{ Li}(\text{g}) + 4 * G \text{ THF}(\text{tol}) + \Delta H \text{ Li}(\text{g} \rightarrow \text{s}) - (S \text{ Li}(\text{s}) - S \text{ Li}(\text{g})) - G [\text{Li}^+(\text{THF})_4][\text{B}^-](\text{tol})$$

$$= -979.2810932 \text{ h} + (-7.4648688 \text{ h}) + 4 * (-231.4634135 \text{ h}) + -159.3 / (627.51 * 4.18) \text{ h} - 298.15 * (29.12 - 138.782) / (627.51 * 4.18 * 1000) \text{ h} - (-1912.6061219 \text{ h})$$

$$= -0.041761 \text{ h} = -26.21 \text{ kcal/mol}$$

Similarly, ΔG revDSD-PBEP86 Complete ionization in toluene was calculated:

$$\Delta G \text{ (kcal/mol)} = G [\text{B}](\text{tol}) + G \text{ Li}(\text{g}) + 4 * G \text{ THF}(\text{tol}) + \Delta H \text{ Li}(\text{g} \rightarrow \text{s}) - (S \text{ Li}(\text{s}) - S \text{ Li}(\text{g})) - G [\text{Li}^+(\text{THF})_4] (\text{tol}) - G [\text{B}^-](\text{tol})$$

$$= -979.2810932 \text{ h} + (-7.4648688 \text{ h}) + 4 * (-231.4634135 \text{ h}) + -159.3 / (627.51 * 4.18) \text{ h} - 298.15 * (29.12 - 138.782) / (627.51 * 4.18 * 1000) \text{ h} - (-933.2591693 \text{ h}) - (-979.3389644 \text{ h})$$

$$= -0.049749 \text{ h} = -31.21 \text{ kcal/mol}$$

Reference:

- [10] Zhao, Y. & Truhlar, D. G. The M06 suite of density functionals for main group thermochemistry, thermochemical kinetics, noncovalent interactions, excited states, and transition elements: two new functionals and systematic testing of four M06-class functionals and 12 other functionals. *Theor. Chem. Acc.* **120**, 215-241 (2008).
- [11] Weigend, F. & Ahlrichs, R. Balanced basis sets of split valence, triple zeta valence and quadruple zeta valence quality for H to Rn: Design and assessment of accuracy. *Phys. Chem. Chem. Phys.* **7**, 3297-3305 (2005).
- [12] Barone, V. & Cossi, M. Quantum Calculation of Molecular Energies and Energy Gradients in Solution by a Conductor Solvent Model. *J. Phys. Chem. A* **102**, 1995-2001 (1998).
- [13] Perdew, J. P., Ruzsinszky, A., Csonka, G. I., Constantin, L. A., Sun, J. Workhorse Semilocal Density Functional for Condensed Matter Physics and Quantum Chemistry. *Phys. Rev. Lett.* **103**, 026403 (2009).
- [14] Jensen, F. Segmented Contracted Basis Sets Optimized for Nuclear Magnetic Shielding. *J. Chem. Theory Comput.* **11**, 132-138 (2015).
- [15] Marenich, A. V., Cramer, C. J., Truhlar, D. G. Universal Solvation Model Based on Solute Electron Density and on a Continuum Model of the Solvent Defined by the Bulk Dielectric Constant and Atomic Surface Tensions. *J. Phys. Chem. B* **113**, 6378-6396 (2009).
- [16] Geuenich, D., Hess, K., Köhler, K., Herges, R. Anisotropy of the Induced Current Density (ACID), a General Method To Quantify and Visualize Electronic Delocalization. *Chem. Rev.* **105**, 3758-3772 (2005).
- [17] Cox, J. D., Wagman, D. D., and Medvedev, V. A., CODATA Key Values for Thermodynamics, Hemisphere Publishing Corp., New York, 1989.
- [18] (a) Santra, G., Sylvetsky, N., Martin, J. M. L., Minimally Empirical Double-Hybrid Functionals Trained against the GMTKN55 Database: revDSD-PBEP86-D4, revDOD-PBE-D4, and DOD-SCAN-D4. *J. Phys. Chem. A* **123**, 5129-5143 (2019) (b) Caldeweyher, E., Bannwarth, C., Grimme, S. Extension of the D3 dispersion coefficient model. *J. Chem. Phys.* **147**, 034112 (2017). (c) Caldeweyher, E., Ehlert, S., Hansen, A., Neugebauer, H., Spicher, S., Bannwarth, C., Grimme, S. A generally applicable atomic-charge dependent London dispersion correction. *J. Chem. Phys.* **150**, 154122 (2019).

Reviewer #2:

This is a very interesting and well-conceived study that adds to the recent flurry of reports that indicate the chemical redox stability of group 1 cations is not as immutable as presented in most textbooks.

In this case the focus is a lithium diphenyl borate in which boron is further ligated by a bipyridine dianion. Although this compound is stable in THF due to coordination of the Li+, dissolution in toluene induces spontaneous reduction to lithium metal and a boron-containing radical; both products are convincingly characterized, by XPS and by X-ray diffraction and EPR, respectively.

Further study of the radical product shows that potassium reduction yields the analogous potassium borate. Although this compound is stable at room temperature, in toluene it undergoes similar self reduction to potassium metal. Furthermore, the potential of this species to act as a two electron reductant, with formation of the boronium cation, is demonstrated through its oxidation with Cu(I). This latter behaviour is then exploited in a sequence of reactions which lead either to reductive E-E coupling of several p-block halides, pyridine and acridine. The potential for such species to act as a

platform for redox-based catalysis is also shown through the reduction of CO₂ to CO by potassium in the presence of 5 mol% the K borate.

This is striking and thought provoking study. While I can think of many questions and suggestions for further experiments (I'm excited to see how general this approach can be made, particularly through extension to alkyl borates and more electron rich bipy variants), the scope of the current work is perfectly suited to publication in Nature Comm and I am, thus, happy to recommend its acceptance.

ANS: We thank the reviewer for the very positive comments.

Minor points:

The introductory section should also make mention of the the recent study of group 1 reduction in Nature Comms 2023, 14, 8147, which will have appeared as this paper was nearing submission.

ANS: We thank the reviewer for the suggestion. The following sentences and literature reference have been added in the revised version of the manuscript.

Page 2, the first paragraph: **The Hill/McMullin groups reported that the reduction of Li⁺/K⁺ can be achieved by other heavier group 1 elements within chloroberyllate species.¹⁷**

17. K. G. Pearce, K. G. et al. Alkali metal reduction of alkali metal cations. *Nature Comm.* 14, 8147 (2023).

Page 3, line 68 should read 'The experimental data were...'

Page 4, Cation to Fig. 2 should read '...in a 4:1 ratio.

ANS: These have been corrected with thanks in the revised version of the manuscript.

Reviewer #3:

This paper was well-revised after submission to other journal where the present reviewer made comments and now it describes chemistry better. However, identification of the present work is still not sufficient, considering the previously reported chemistry of bipyridines were not adequately referred and discussion on aromaticity is completely wrong. Therefore, the reviewer does not agree with publishing this paper. Please find the comments.

1) The largest claim in this paper is reducing ability of compound 5. However, this phenomena comes from the redox property of bipyridine. The reduced form of bipyridine at the coordination sphere of

early transition metals has been reported. These examples should be cited.
<https://pubs.acs.org/doi/10.1021/acs.inorgchem.6b02954>
<https://chemistry-europe.onlinelibrary.wiley.com/doi/10.1002/chem.202003465>
<https://pubs.acs.org/doi/10.1021/om900943m>
<https://www.sciencedirect.com/science/article/pii/S0022328X1000505X?via=ihub>
<https://pubs.acs.org/doi/10.1021/acs.inorgchem.7b01344>
<https://pubs.rsc.org/en/content/articlelanding/2018/SC/C7SC05238C>

ANS: We thank the reviewer's suggestion. The following sentences and literature references have been added in the revised version of the manuscript.

Page 3, the first paragraph: Similarly, the reduced form of bipyridine at the coordination sphere of early transition metals has been reported.²⁰⁻²⁵

20. Rosenzweig, M. W. et al. Molecular and Electronic Structures of Eight-Coordinate Uranium Bipyridine Complexes: A Rare Example of a Bipy2– Ligand Coordinated to a U⁴⁺ Ion. *Inorg. Chem.* **56**, 2792–2800 (2017).
21. Wang, D. et al. Experimental and Computational Studies on a Base-Free Terminal Uranium Phosphinidene Metallocene, *Chem. Eur. J.* **26**, 16888–16899 (2020).
22. Carver, C. T. et al. Coupling of Aromatic N-Heterocycles Mediated by Group 3 Complexes, *Organometallics* **29**, 835–846 (2010).
23. Yahia, A. et al. C-C coupling reaction of pyridine derivatives at the dimethyl rare-earth metal cation [YMe₂(THF)₃]⁺: A DFT investigation, *J. Organometallic Chem.* **695**, 2789–2793 (2010).
24. Kurogi, T. et al. Room temperature olefination of methane with titanium–carbon multiple bonds, *Chem. Sci.*, **9**, 3376–3385 (2018).
25. Fedushkin, I. L. et al. Ytterbium and Europium Complexes of Redox-Active Ligands: Searching for Redox Isomerism, *Inorg. Chem.* **56**, 9825–9833 (2017).

Also, electron transfer from naphthalene radical anion has been used in organic synthesis for a long time. Such chemistry should be compared with reactivity of 5 and 6 in this paper, especially for Schemes 6-9.

ANS: We thank the reviewer's suggestion. we conducted the experiments as shown in Schemes 6-8, using naphthalene anion as the reducing reagent. However, only the P-P coupling reaction can be realized with a 21% conversion, and the other reactions can't give the desired products. The protonation of naphthalene anion was known, giving dihydronaphthalene as the final product (THE REACTIONS OF SODIUM NAPHTHALENIDE WITH CARBONYL COMPOUNDS AND ESTERS, Master thesis, Mannar M. Vora, **1972**). For CO₂ reduction, it was reported that naphthalene anion could be transferred into 1,4-dicarboxylated 1,4-dihydronaphthalene derivatives (*J. Am. Chem. Soc.* **1959**, *81*, 2067–2069; *Org. Lett.* **2023**, *25*, 4231–4235). We have added the following sentences in the revised version of the manuscript and supporting information.

Page 8, the second paragraph: In comparison, when using sodium naphthalene as the reducing reagent, a 21% generation of P₂Ph₄ was observed based on the ³¹P NMR (See ESI, Figure S48).

Page 9, the first paragraph: When using sodium naphthalene as the reducing reagents in these reactions, the coupling products can't be obtained.

Page 9, the second paragraph: These two reactions can't be realized using sodium naphthalene as a reducing reagent.

Supporting information:

Reactions of Scheme 6-8 using naphthalene anion as reducing reagents

Sodium naphthalene was freshly prepared from a mixture of naphthalene and sodium in THF at -35°C (concentration was 0.1 mol/L). For E-E reactions, only P-P was formed with a 21% conversion. The Sn-Sn, Se-Se, Ge-Ge coupling products were not observed. In the reductive coupling of pyridine, the formation of 4,4'-bipyridine was not observed. In the Birch reduction of Acridine, no desired products were observed.

Sn-Sn coupling reactions:

To the C_6D_6 solution of $n\text{Bu}_3\text{SnCl}$ (6.5 mg, 0.02 mmol, 1.0 eq.) in J. Young's tube was added 200 μL 0.1M Sodium naphthalene (0.02 mmol, 1.0eq.). The reaction was placed at room temperature overnight. No Sn-Sn products were observed based on the $^1\text{H NMR}$.

Figure S45. $^1\text{H NMR}$ in C_6D_6 for comparison (Sn-Sn). (Red: $n\text{Bu}_3\text{SnCl} + \text{NaNap.}$; Green: $n\text{Bu}_6\text{Sn}_2$).

Se-Se coupling reactions:

To the C_6D_6 solution of PhSeCl (3.8 mg, 0.02 mmol, 1.0 eq.) in J. Young's tube was added 200 μL 0.1M Sodium naphthalene (0.02 mmol, 1.0eq.). The reaction was placed at room temperature overnight. No generation of Se-Se compounds was observed based on the $^1\text{H NMR}$.

Figure S46. ^1H NMR in C_6D_6 for comparison (Ge-Ge). (Red: $\text{PhSeCl} + \text{NaNap.}$; Green: Ph_2Se_2).

Ge-Ge coupling reactions:

To the C_6D_6 solution of Et_3GeCl (3.9 mg, 0.02 mmol, 1.0 eq.) in J. Young's tube was added 200 μ L 0.1M Sodium naphthalene (0.02 mmol, 1.0eq.). The reaction was placed at room temperature overnight. No generation of Ge-Ge compounds was observed based on the 1H NMR.

Figure S47. 1H NMR in C_6D_6 for comparison (Ge-Ge). (Red: Et_3GeCl ; Green: Et_3GeCl + NaNap., removal of solvent).

P-P coupling reactions:

To the C_6D_6 solution of Ph_2PCl (4.4 mg, 0.02 mmol, 1.0 eq.) in J. Young's tube was added 200 μ L 0.1M Sodium naphthalene (0.02 mmol, 1.0eq.). The reaction was placed at room temperature overnight. About 21% of P-P compounds were generated based on the ^{31}P NMR.

Figure S48. $^{31}P\{^1H\}$ NMR in C_6D_6 for comparison (P-P).

Reductive Pyridine couplings:

To 0.1M Sodium naphthalene (0.63 mmol, 1.0 eq.) was added pyridine (50 mg, 0.63 mmol, 1.0 eq.). After stirred at room temperature overnight. drops of MeOH were added to quench the reaction. No 4,4'-bipyridine was observed based on TLC (EA: DCM=2:1, 1%NEt₃).

The reduction of acridine:

To 0.1M Sodium naphthalene (0.28 mmol, 1.0 eq.) was added acridine (50 mg, 0.28 mmol, 1.0 eq.). After stirring at room temperature overnight. drops of MeOH were added to quench the reaction. The reduction product was not observed on TLC (EA: Hex=1:5).

2) As mentioned in the previous reviewing, the statement "It is found that the BN2C2 rings in 5 and 6 are aromatic with negative NICS(1) values, whereas the two C5N rings are dearomatized with positive NICS(1) values, especially for the borate anion 5, showing NICS(1) values of 25.89 and 25.92 (Figure 6)." in page 7 is wrong, considering the BN2C2 ring is not planar. The authors should know the NICS values are chemical shift and it can be affected by a change in charge density. As the authors added ACID plot in SI after the previous reviewing process, however, there is no ring current over B-N bonds, suggesting BN2C2 ring is not aromatic.

ANS: We agree that it is not appropriate to discuss the aromaticity of the BN2C2 ring, since the ring is not planar. We have corrected the sentences in the revised version of the manuscript.

Page 7, the first paragraph: In structures 5 and 6, the two C5N rings are dearomatized with positive NICS(1) values, especially for the borate anion 5, showing NICS(1) values of 25.89 and 25.92 (Figure 6). In comparison, the two C5N rings in 7 are aromatic and exhibit negative NICS(1) values (-23.32). The extent of π -electron delocalization in 5-7 was further assessed through the analysis of current-induced density (ACID) anisotropy (see ESI, Figure S42-S44).

Reviewers' Comments:

Reviewer #1:

Remarks to the Author:

All concerns raised by the reviewers, including mine, have been addressed in the revision. The formation of lithium metal is now supported by theoretical calculations, and the uniqueness of the bipyridyl-containing borate anion as a two-electron reductant has been verified by conducting reactions with sodium naphthalene. With these modifications implemented, I fully support the publication of the revised manuscript in Nature Communications.

Reviewer #3:

Remarks to the Author:

After the reviewing process, the paper seems to be appropriately revised to solve the scientific conflict. Therefore, the reviewer is happy to recommend publishing the paper in Nature Communications.